# R-spondins can potentiate WNT signaling without LGRs

**Andres M Lebensohn[1,2]\*, Rajat Rohatgi[1,2]\***

[1]Department of Biochemistry, Stanford University School of Medicine, Stanford, United States; [2]Department of Medicine, Stanford University School of Medicine, Stanford, United States

**Abstract** The WNT signaling pathway regulates patterning and morphogenesis during development and promotes tissue renewal and regeneration in adults. The R-spondin (RSPO) family of four secreted proteins, RSPO1-4, amplifies target cell sensitivity to WNT ligands by increasing WNT receptor levels. Leucine-rich repeat-containing G-protein coupled receptors (LGRs) 4-6 are considered obligate high-affinity receptors for RSPOs. We discovered that RSPO2 and RSPO3, but not RSPO1 or RSPO4, can potentiate WNT/β-catenin signaling in the absence of all three LGRs. By mapping the domains on RSPO3 that are necessary and sufficient for this activity, we show that the requirement for LGRs is dictated by the interaction between RSPOs and the ZNRF3/RNF43 E3 ubiquitin ligases and that LGR-independent signaling depends on heparan sulfate proteoglycans (HSPGs). We propose that RSPOs can potentiate WNT signals through distinct mechanisms that differ in their use of either LGRs or HSPGs, with implications for understanding their biological functions.

DOI: https://doi.org/10.7554/eLife.33126.001

**\*For correspondence:**
andreslebensohnphd@gmail.com
(AML);
rrohatgi@stanford.edu (RR)

**Competing interests:** The authors declare that no competing interests exist.

## Introduction

The WNT signaling pathway regulates patterning and morphogenesis during development (*Hoppler and Moon, 2014*). In adults, WNT ligands promote the renewal of stem cells, maintaining tissue homeostasis during routine turnover or following injury (*Clevers et al., 2014*). Some WNT responses in vertebrates depend on a second signal provided by the R-spondin (RSPO) family of four secreted proteins (*de Lau et al., 2012*). RSPOs have emerged as important stroma-derived growth factors that drive the renewal of epithelial stem cells in many adult vertebrate tissues (*de Lau et al., 2014*). RSPOs markedly amplify target cell sensitivity to WNT ligands by neutralizing two transmembrane E3 ubiquitin ligases, ZNRF3 and RNF43, which reduce cell-surface levels of WNT receptors (*Hao et al., 2012*; *Koo et al., 2012*). Indeed, chromosomal translocations that increase RSPO expression or mutations that inactivate ZNRF3/RNF43 can drive cancer (*Hao et al., 2016*).

RSPOs contain tandem furin-like repeats (FU1 and FU2), a thrombospondin type 1 (TSP) domain and a basic region (BR). RSPOs simultaneously engage ZNRF3/RNF43 through their FU1 domain and leucine-rich repeat-containing G-protein coupled receptors (LGRs) 4-6 through their FU2 domain (*Chen et al., 2013*; *Peng et al., 2013*; *Wang et al., 2013*; *Xu et al., 2013*; *Zebisch et al., 2013*), triggering the clearance of ZNRF3/RNF43 and the consequent rise in WNT receptor levels on the cell surface. LGR4-6 are selectively expressed in various tissue stem cells (*Clevers et al., 2014*) and are considered the primary high-affinity receptors for RSPOs (*Carmon et al., 2011*; *de Lau et al., 2011*; *Glinka et al., 2011*).

We now show that RSPO2 and RSPO3, but not RSPO1 or RSPO4, can potentiate WNT/β-catenin signaling in the absence of all three LGRs. Using purified mutant and chimeric RSPOs and cell lines lacking various receptors, we elucidated the requirements for this mode of signaling. The ZNRF3/RNF43-interacting FU1 domain is necessary for LGR-independent signaling, while the LGR-

interacting FU2 domain is dispensable. The FU1 domain of RSPO3 is also sufficient to confer the capacity to signal without LGRs when transplanted to RSPO1. The TSP/BR domains of RSPOs and their interaction with heparan sulfate proteoglycans (HSPGs), previously considered dispensable for potentiation of WNT/β-catenin signaling (*Kazanskaya et al., 2004*; *Ohkawara et al., 2011*), are essential in the absence of LGRs. These results define two alternative modes of RSPO-mediated signaling that share a common dependence on ZNRF3/RNF43, but differ in their use of either LGRs or HSPGs.

## Results

In previous work (*Lebensohn et al., 2016*), we generated and thoroughly characterized a haploid human cell line (HAP1-7TGP) that harbors a fluorescent transcriptional reporter for WNT/β-catenin signaling. Both the fluorescence of this synthetic reporter and the transcription of endogenous WNT target genes in HAP1-7TGP cells can be activated by WNT ligands and these WNT responses can be strongly potentiated by RSPOs (*Lebensohn et al., 2016*). HAP1-7TGP cells do not secrete WNT ligands and thus their response to RSPOs requires the co-administration of a low concentration of WNT. A comprehensive set of unbiased genetic screens in HAP1-7TGP cells identified many of the known components required for a signaling response to WNT and RSPO ligands, establishing this cell line as a valid and genetically tractable system to study this pathway (*Lebensohn et al., 2016*).

We made the serendipitous and unexpected observation that RSPO3 could potently enhance WNT reporter fluorescence driven by a low concentration of WNT3A in two independently derived HAP1-7TGP clonal cell lines carrying loss-of-function mutations in *LGR4* (LGR4$^{KO}$ cells; see Materials and methods and *Supplementary file 1*) (*Figure 1A*). In contrast, RSPO1 did not enhance the response to WNT3A in LGR4$^{KO}$ cells. RSPO1 and RSPO3 had equivalent activity in wild-type (WT) HAP1-7TGP cells, demonstrating that both ligands were functional, and the response to RSPO3 in both WT and LGR4$^{KO}$ cells depended on the presence of WNT3A (*Figure 1A*).

While *LGR4* is the only RSPO receptor expressed in HAP1 cells (*Table 1*) (*Dubey et al., 2016*), we excluded the possibility of compensatory up-regulation of *LGR5* or *LGR6* by simultaneously disrupting both genes in LGR4$^{KO}$ cells, generating multiple independent clonal cell lines lacking all three RSPO receptors (hereafter called LGR4/5/6$^{KO}$ cells; *Supplementary file 1*). LGR4/5/6$^{KO}$ cells retained an intact WNT signaling cascade, responding normally to a saturating dose of WNT3A (*Figure 1B*). All four RSPOs (1-4) strongly potentiated WNT signaling in WT cells, establishing ligand integrity. However, RSPO1 and RSPO4 were completely inactive in LGR4/5/6$^{KO}$ cells, even at concentrations that induced maximum WNT reporter induction in WT cells, whereas RSPO2 and RSPO3 strongly potentiated signaling driven by low concentrations of WNT3A in the absence of all three LGRs (*Figure 1B*). Therefore, RSPO2 and RSPO3 possess a unique quality absent in RSPO1 and RSPO4 that enables them to potentiate WNT responses without LGRs.

Dose-response analysis revealed that RSPO1 and RSPO3 enhanced WNT signaling in WT cells with nearly identical pharmacodynamics—both the efficacy (maximum reporter activity) and the potency (measured by the EC50, defined as the RSPO concentration that induced half-maximum reporter activity) were similar for both ligands (*Figure 1C*). In LGR4/5/6$^{KO}$ cells, RSPO1 had no detectable activity at concentrations up to 160 ng/ml, 400-fold higher than its EC50 in WT cells. While RSPO3 potentiated WNT signaling in LGR4/5/6$^{KO}$ cells, its efficacy was reduced by 33% and its EC50 was increased by 16-fold compared to WT cells (*Figure 1C*). The distinct pharmacodynamics of RSPO3 in the two cell types suggested that its reception was mediated by different mechanisms in the presence and absence of LGRs.

We sought to determine which domains of RSPO3 were required for LGR-independent signaling using a ligand mutagenesis strategy (*Figure 2A*). Our experimental strategy leveraged a comparison between RSPO1 and RSPO3, since the former depended strictly on LGRs while the latter could signal in their absence. Unless otherwise noted, each WT and mutant RSPO ligand described hereafter was produced as a fusion protein carrying an N-terminal hemagglutinin (HA) tag and a tandem C-terminal tag composed of an immunoglobulin fragment crystallizable (Fc) domain followed by a 1D4 epitope tag (*Molday and Molday, 2014*) used for immuno-affinity purification (see Materials and methods and *Figure 1—figure supplement 1A and B* for a description of ligand purification and characterization). Importantly, the pharmacodynamics of the tagged RSPO proteins were similar to

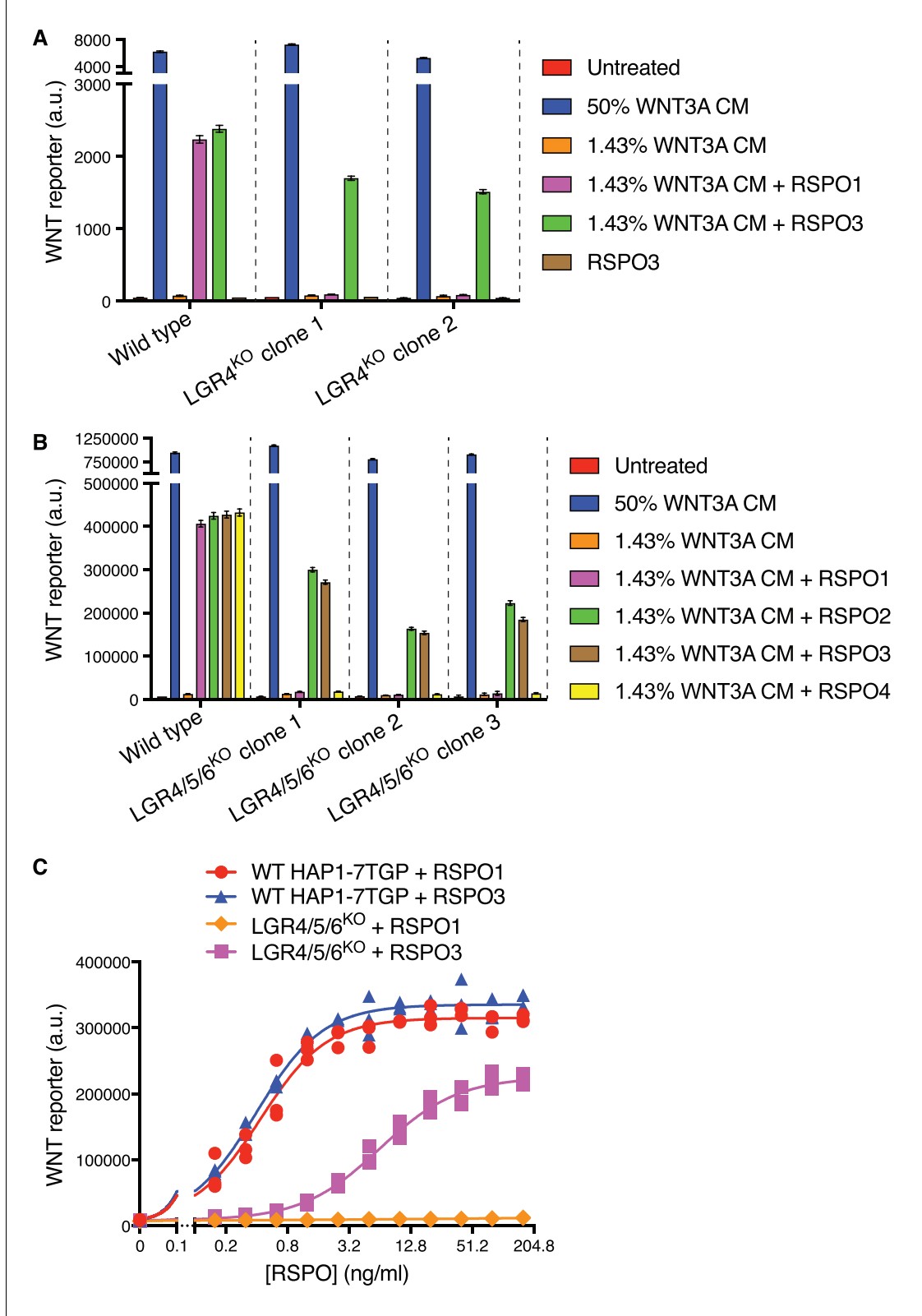

**Figure 1.** RSPO2 and RSPO3 can potentiate WNT signaling in the absence of LGR4, LGR5 and LGR6. (**A**) WNT reporter fluorescence (median ±standard error of the median (SEM) from 10,000 cells) for WT HAP1-7TGP and two distinct LGR4$^{KO}$ clonal cell lines (determined by genotyping, *Supplementary file 1*) following treatment with the indicated combinations of WNT3A conditioned media (CM) and untagged, recombinant RSPO1 or RSPO3 (both at 20 ng/ml). All cell lines responded similarly to a saturating dose of WNT3A, demonstrating an intact downstream signaling response. (**B**) *Figure 1 continued on next page*

*Figure 1 continued*

WNT reporter fluorescence (median ±SEM from 10,000 cells) for WT HAP1-7TGP and three distinct LGR4/5/6[KO] clonal cell lines (*Supplementary file 1*) treated with the indicated combinations of WNT3A CM and various RSPOs. RSPO1, RSPO2 and RSPO3 were used at 40 ng/ml and RSPO4 at 400 ng/ml, concentrations that produced equivalent responses in WT cells. (C) Dose-response curves for RSPO1 and RSPO3 in WT HAP1-7TGP and LGR4/5/6[KO] cells in the presence of 1.43% WNT3A CM. Each symbol represents the median WNT reporter fluorescence from a single well and measurements from three independently treated wells are shown for each RSPO concentration. The curves were fitted as described in Materials and methods.

DOI: https://doi.org/10.7554/eLife.33126.002

The following figure supplement is available for figure 1:

**Figure supplement 1.** Immuno-affinity purification and functional characterization of recombinant, tagged RSPO proteins used in this study.

DOI: https://doi.org/10.7554/eLife.33126.003

those of their untagged counterparts in both WT and LGR4/5/6[KO] cells (*Figure 1—figure supplement 1C and D*).

Previous studies have shown that the FU1 and FU2 domains in all RSPOs, which bind to ZNRF3/RNF43 and LGRs, respectively, are both necessary and sufficient to potentiate WNT responses, while the TSP and BR domains are dispensable (*Kazanskaya et al., 2004*). Simultaneous deletion of the FU1 and FU2 domains of RSPO3 abolished signaling in both WT and LGR4/5/6[KO] cells (*Figure 2B*). Point mutations in the FU1 domain of RSPO3 (R67A/Q72A; *Figure 2A*) known to weaken the interaction between RSPOs and ZNRF3/RNF43 (*Xie et al., 2013*) entirely abolished RSPO3 signaling in LGR4/5/6[KO] cells (*Figure 2B and D*) and substantially reduced (but did not abolish) RSPO3 signaling in WT cells (EC50 increased by 21-fold; *Figure 2B and C*). Thus, the reduction in the affinity between RSPO3 and ZNRF3/RNF43 caused by the FU1 R67A/Q72A mutation impaired LGR-independent signaling to a much greater extent than LGR-dependent signaling. Indeed, the equivalent R66A/Q71A mutation in RSPO1 (*Figure 2A*), which only signals in an LGR-dependent manner, also impaired but did not completely abolish signaling in WT cells (*Figure 2B*).

Point mutations in the FU2 domain of RSPO3 (F106E/F110E; *Figure 2A*) that weaken interactions with LGRs (*Xie et al., 2013*) had little impact on RSPO3 signaling in LGR4/5/6[KO] cells, consistent with the lack of LGRs in these cells (*Figure 2B and D*). In WT cells, the F106E/F110E mutation in RSPO3 did not prevent signaling, but reduced the efficacy by 48% and increased the EC50 by 2.9-fold (*Figure 2C*). Thus, RSPO3 signaling in WT cells includes contributions from both LGR-dependent and independent pathways. In contrast, the F106E/F110E mutation in RSPO1 abolished signaling in WT cells, demonstrating that signaling by RSPO1 depends entirely on its interaction with LGRs (*Figure 2B*).

The C-terminal TSP and BR domains of RSPOs (denoted TSP/BR when discussed together) are considered dispensable for LGR-mediated signaling (*Glinka et al., 2011*). When we deleted these domains individually in RSPO3, there were only minor effects on signaling in WT cells (*Figure 2E and F*). Deletion of both domains in RSPO3

**Table 1.** Relative gene expression level in HAP1 cells of selected genes discussed in this work. RPKM (Reads Per Kilobase of transcript per Million mapped reads) values from duplicate RNA-seq datasets described previously (NCBI GEO accession number GSE75515, https://www.ncbi.nlm.nih.gov/geo/), obtained from two different passages of WT HAP1 cells (*Dubey et al., 2016*). Groups of paralogues or genes with redundant function are shaded in alternating colors to facilitate comparisons.

| Gene | RPKM | | |
| | Replicate 1 | Replicate 2 | Average |
|------|-------------|-------------|---------|
| LGR4 | 160.61 | 174.69 | 167.65 |
| LGR5 | 0.02 | 0.00 | 0.01 |
| LGR6 | 0.02 | 0.00 | 0.01 |
| ZNRF3 | 30.9 | 33.3 | 32.1 |
| RNF43 | 0.12 | 0.08 | 0.1 |
| GPC1 | 49.55 | 47.53 | 48.54 |
| GPC2 | 4.17 | 4.79 | 4.48 |
| GPC3 | 170.22 | 144.37 | 157.29 |
| GPC4 | 209.39 | 229.86 | 219.63 |
| GPC5 | 0.1 | 0.1 | 0.1 |
| GPC6 | 13.88 | 14.90 | 14.39 |
| SDC1 | 51.37 | 47.88 | 49.63 |
| SDC2 | 11.42 | 9.2 | 10.31 |
| SDC3 | 43.58 | 50.64 | 47.11 |
| SDC4 | 8.16 | 8.21 | 8.18 |

DOI: https://doi.org/10.7554/eLife.33126.004

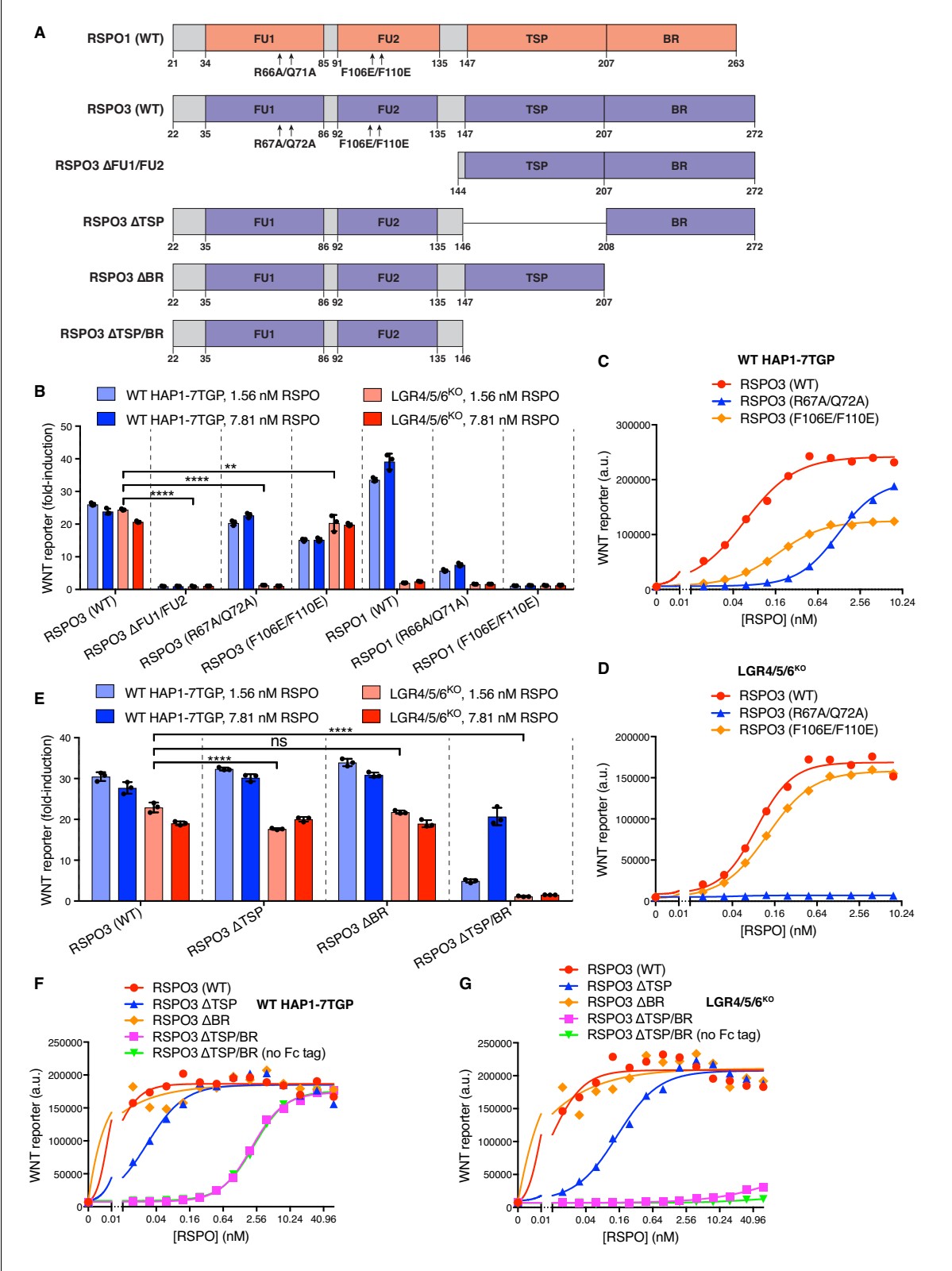

**Figure 2.** Domains of RSPO3 required for LGR-independent signaling. (**A**) Schematic representation of human WT and mutant RSPO1 (salmon) and RSPO3 (violet) proteins produced and purified as described in Materials and methods and *Figure 1—figure supplement 1A*. The N-terminal HA and C-terminal Fc and 1D4 tags present in all constructs are not shown. Amino acid numbers for human RSPO1 and RSPO3 (UniProt accession number Q2MKA7 and Q9BXY4, respectively) are indicated below and arrows show mutations made in the FU1 and FU2 domains. Polypeptide lengths are drawn

*Figure 2 continued on next page*

*Figure 2 continued*

to scale. (B and E) Fold-induction in WNT reporter fluorescence (over 1.43% WNT3A CM alone) caused by treatment of WT HAP1-7TGP (light blue and blue bars) and LGR4/5/6$^{KO}$ (salmon and red bars) cells with two concentrations of purified RSPO proteins. Each circle indicates the fold-induction for a single well, calculated as the median WNT reporter fluorescence from each well treated with 1.43% WNT3A CM and RSPO, divided by the average of the median WNT reporter fluorescence from triplicate wells treated with 1.43% WNT3A CM alone. Bars and error bars indicate the average ±SD of triplicate wells. Significance was determined as described in Materials and methods. (C, D, F, G) Dose-response curves for the indicated purified RSPO proteins in WT HAP1-7TGP (C, F) and LGR4/5/6$^{KO}$ (D, G) cells, in the presence of 1.43% WNT3A CM. Each symbol represents the median WNT reporter fluorescence from 5000 cells. In F and G, RSPO3 ΔTSP/BR was tested with and without the dimerizing Fc tag.

DOI: https://doi.org/10.7554/eLife.33126.005

increased the EC50 in WT cells by 333-fold, but did not change the efficacy (*Figure 2F*). Therefore, while the TSP/BR domains are not essential for signaling in WT cells, consistent with prior work, their loss substantially reduces the potency of RSPO3. In contrast, RSPO3 lacking the TSP/BR domains failed to potentiate WNT responses in LGR4/5/6$^{KO}$ cells at the concentrations tested (*Figure 2E and G*). The signaling properties of RSPO3 lacking the TSP/BR domains were unchanged when the dimerizing Fc tag was removed (*Figure 2F and G*).

These mutagenesis experiments demonstrated that the FU1 and TSP/BR domains of RSPO3 are required for its ability to potentiate WNT responses in the absence of LGRs, while the FU2 domain is dispensable. These domain requirements are distinct from those for LGR-mediated signaling by RSPO1, which depends on the FU1 and FU2, but not on the TSP/BR domains. In WT cells, RSPO3 signaling proceeded through both LGR-dependent and independent mechanisms because disruption of the FU2 or the TSP/BR domains partially impaired but did not abolish signaling (*Figure 2C and F*). The ZNRF3/RNF43-interacting FU1 domain is essential for signaling both in the presence and absence of LGRs.

To identify the region of RSPO3 that confers the capacity to signal without LGRs, we constructed a series of chimeric ligands by combining regions of RSPO1 and RSPO3 (*Figure 3A*). Remarkably, replacing the FU1 domain of RSPO1 with the FU1 domain of RSPO3 enabled RSPO1 to potentiate WNT signaling in LGR4/5/6$^{KO}$ cells (*Figure 3B and D*). Conversely, replacing the FU1 domain of RSPO3 with that of RSPO1 drastically reduced the signaling capacity of RSPO3 in LGR4/5/6$^{KO}$ cells (*Figure 3B and D*). In important control experiments, all chimeric ligands showed equivalent activity in WT cells, establishing ligand integrity (*Figure 3B and C*). Thus, a difference in the interaction between ZNRF3/RNF43 and the FU1 domains of RSPO1 and RSPO3 is the crucial determinant of the requirement for LGRs. Of note, the affinities of the FU1-FU2 fragment of RSPO2 (25 nM) and RSPO3 (60 nM) for ZNRF3 have been reported to be much higher than those of RSPO1 (6.8 μM) and RSPO4 (300 μM) (*Zebisch et al., 2013*). Indeed, these affinities correlate with the capacity of RSPO2 and RSPO3, but not RSPO1 or RSPO4, to promote LGR-independent signaling (*Figure 1B*). While the TSP/BR domains of RSPO3 were required for LGR-independent signaling, they were not sufficient because replacing the TSP/BR domains of RSPO1 with those of RSPO3 did not confer on RSPO1 the capacity to signal in LGR4/5/6$^{KO}$ cells (*Figure 3E*). In fact, the TSP/BR domains of RSPO1 and RSPO3 seemed interchangeable for signaling activity in both WT and LGR4/5/6$^{KO}$ cells (*Figure 3E*).

These results suggested that the WNT-potentiating activity of RSPO3 in the absence of LGRs depends on its interaction with ZNRF3/RNF43 through the FU1 domain and additional interactions with an alternative co-receptor through the TSP/BR domains. We considered the previous observation that the TSP/BR domains of RSPOs can bind to heparin (*Nam et al., 2006*). Addition of heparin to the culture medium completely blocked potentiation of WNT signaling by RSPO3 in LGR4/5/6$^{KO}$ cells, but had only a partial inhibitory effect in WT cells, in which RSPO3 can also signal through LGRs (*Figure 4A*).

The TSP/BR domains of RSPOs can mediate interactions with the two major families of cell-surface HSPGs, the glycophosphatidylinositol (GPI)-linked glypicans and the transmembrane syndecans (*Ohkawara et al., 2011*). In humans, both protein families are encoded by multiple, partially redundant genes: six glypican genes (*GPC1-6*) and four syndecan genes (*SDC1-4*) (*Park et al., 2000*), all of which are expressed in HAP1 cells except for *GPC5* (*Table 1*). Since all glypicans and syndecans must be post-translationally modified with heparan sulfate chains for receptor function, we disrupted *EXTL3*, a gene encoding a glycosyltransferase that is specifically required for HSPG biosynthesis, but dispensable for the synthesis of other glycosaminoglycans and proteoglycans (*Takahashi et al.,*

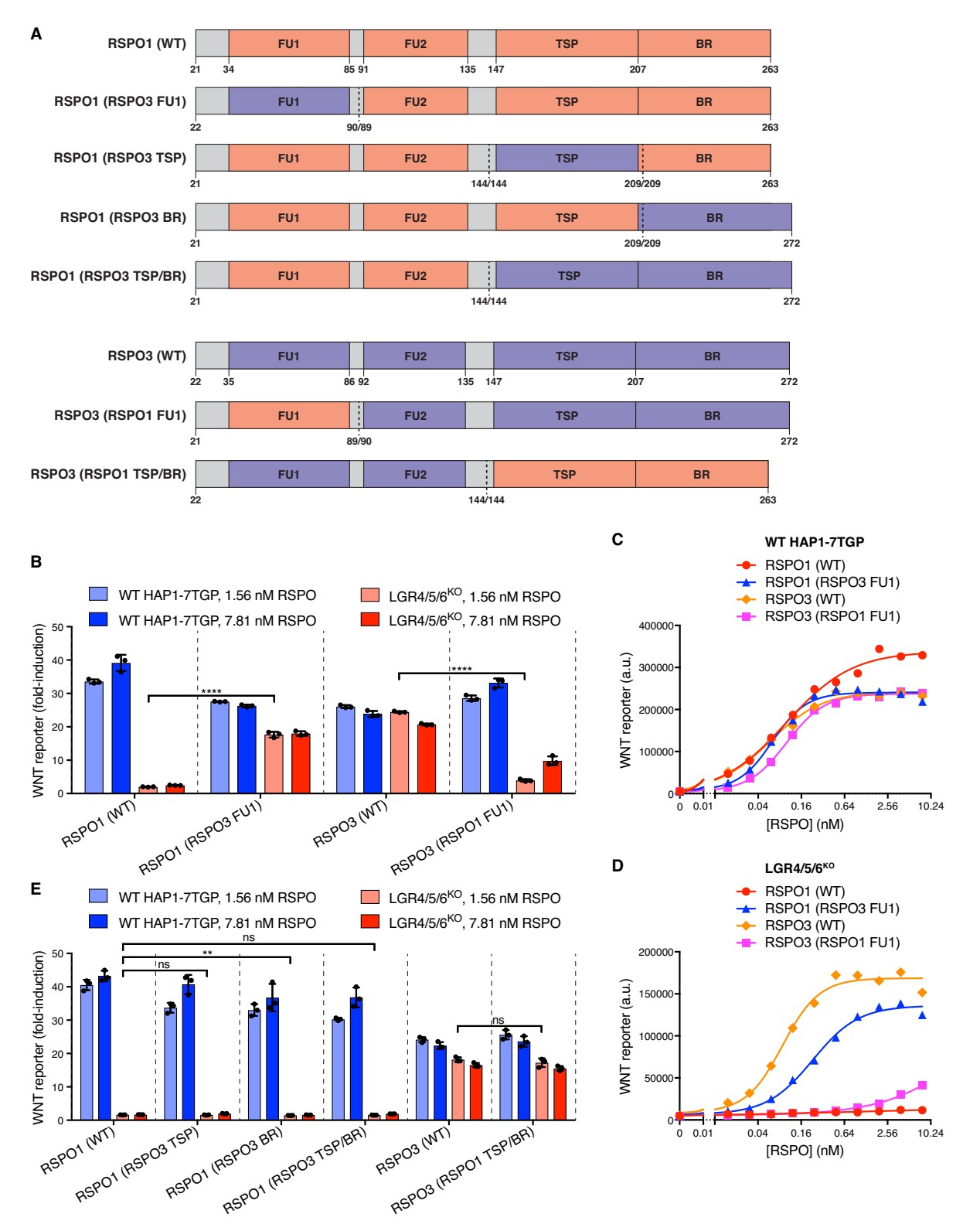

**Figure 3.** Domains of RSPO3 sufficient to confer the capacity to signal without LGRs. (**A**) Schematic representation of human WT and chimeric RSPO1 (salmon) and RSPO3 (violet) proteins, depicted as in *Figure 2A*. Vertical dotted lines indicate the sites at which swaps were made. Each swap was made at a conserved amino acid indicated under each construct (the amino acid numbers on the left and right of the slash correspond to the proteins depicted on the left and right of the dotted line, respectively). (**B and E**) Fold-induction in WNT reporter fluorescence (over 1.43% WNT3 CM alone) as
*Figure 3 continued on next page*

Figure 3 continued

described in **Figure 2B, E**. (**C and D**) Dose-response curves for the indicated purified RSPO proteins in WT HAP1-7TGP (**C**) and LGR4/5/6$^{KO}$ (**D**) cells, in the presence of 1.43% WNT3A CM. Each symbol represents the median WNT reporter fluorescence from 5000 cells.

DOI: https://doi.org/10.7554/eLife.33126.006

*2009*). The loss of EXTL3 in LGR4/5/6$^{KO}$ cells led to an 81% reduction in RSPO3-mediated potentiation of WNT signaling (*Figure 4B*). In contrast, the loss of EXTL3 in WT cells only reduced signaling by 34%, likely because RSPO3 can also signal through LGRs in WT cells. In an important control, the loss of EXTL3 did not affect signaling induced by addition of a sub-saturating concentration of WNT3A alone or by inhibition of the β-catenin destruction complex kinase GSK3 in either WT or LGR4/5/6$^{KO}$ cells (*Figure 4—figure supplement 1A*).

To distinguish between glypicans and syndecans, we took advantage of the fact that only glypicans are anchored to the cell surface by a GPI linkage. Disrupting *PIGL*, a gene required for GPI-anchor biosynthesis, or disrupting both *GPC4* and *GPC6*, the two glypican genes identified in our previous haploid genetic screens (*Lebensohn et al., 2016*), in LGR4/5/6$^{KO}$ cells did not impair LGR-independent potentiation of WNT signaling by RSPO3 (*Figure 4C*). As we had found previously in WT HAP1-7TGP cells (*Lebensohn et al., 2016*), disrupting *PIGL* or *GPC4* and *GPC6* in LGR4/5/6$^{KO}$ cells reduced signaling by a sub-saturating dose of WNT3A (*Figure 4—figure supplement 1B*), indicating that glypicans are required to mediate cellular responses to low doses of WNT3A. Disrupting all four syndecans (*SDC1-4*) simultaneously in LGR4/5/6$^{KO}$ cells did not impair LGR-independent potentiation of WNT signaling by RSPO3 (*Figure 4D*), nor signaling induced by sub-saturating WNT3A or by inhibition of GSK3 (*Figure 4—figure supplement 1C*).

We conclude from these results that either glypicans or syndecans (or another HSPG) can mediate LGR-independent signaling, probably through interactions between their heparan sulfate chains and the TSP/BR domains of RSPO3. Therefore, disrupting any single family of HSPGs does not impair signaling (*Figure 4C and D*), while disrupting EXTL3, an enzyme required for the biosynthesis of heparan sulfate chains present in all HSPGs, substantially reduces LGR-independent signaling (*Figure 4B*).

## Discussion

Our study shows that RSPOs can potentiate WNT signals in the absence of LGRs, expression of which has been hitherto considered the hallmark of RSPO-responsive cells. This quality is unique to RSPO2 and RSPO3 and is endowed by their ZNRF3/RNF43-interacting FU1 domain, since transplanting the FU1 domain from RSPO3 renders RSPO1 capable of signaling in the absence of LGRs (*Figure 3B, D*). Furthermore, the TSP/BR domains of RSPO3, which had been deemed dispensable for LGR-dependent signaling (*Glinka et al., 2011*), are necessary for signaling in the absence of LGRs (*Figure 2E, G*). HSPGs, cell surface proteins capable of interacting with the TSP/BR domains (*Ohkawara et al., 2011*), are also selectively required for RSPO3-dependent potentiation of WNT signaling in the absence of LGRs (*Figure 4B*). Thus, our results suggest that the interaction of the TSP/BR domains of RSPO3 with HSPGs provides an alternative mechanism that neutralizes ZNRF3/RNF43 in the absence of LGRs (*Figure 4E*). HSPGs are known to mediate the efficient endocytosis of multiple cargoes (*Christianson and Belting, 2014*). Hence, we speculate that the simultaneous interaction of RSPO3 with ZNRF3/RNF43 through its FU1 domain and HSPGs through its TSP/BR domains provides an LGR-independent route for the endocytosis and clearance of ZNRF3/RNF43 from the cell surface (*Figure 4E*) and the consequent rise in WNT receptor levels.

Future work will be necessary to define the developmental, regenerative or oncogenic contexts in which this LGR-independent mode of signaling is used to amplify target cell responses to WNT ligands. The presence of three LGRs capable of mediating responses to four RSPO ligands (*de Lau et al., 2011*) poses significant experimental challenges to addressing this question genetically in mouse models, as we did in human haploid cells, since multiple genes would need to be simultaneously disrupted to test if any responses to RSPOs are retained in tissues lacking all three LGRs.

However, some information can be gleaned from the developmental phenotypes of mice carrying mutations in *Rspo2* or *Rspo3*, the two members of the family capable of potentiating WNT signaling in cells lacking all three LGRs (*Figure 1B*). *Rspo2* is expressed in the limb and lung buds of the early

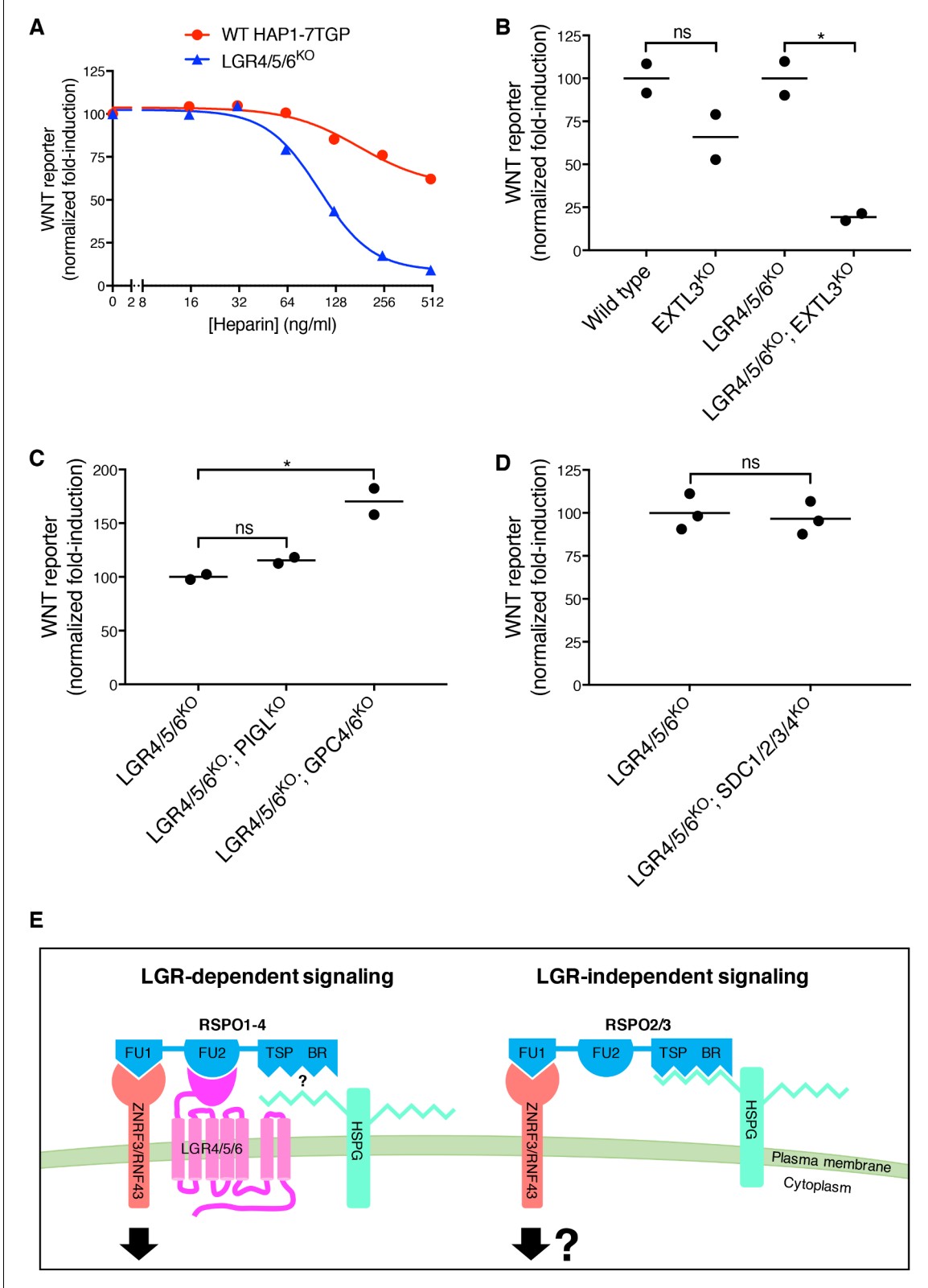

**Figure 4.** LGR-independent signaling by RSPO3 requires heparan sulfate proteoglycans. (**A**) WNT reporter induction in WT HAP1-7TGP and LGR4/5/6[KO] cells treated with 1.43% WNT3A CM, 2 nM untagged RSPO3 and the indicated concentrations of heparin. For each heparin concentration, the fold-induction (over 1.43% WNT3A CM alone) by RSPO3 was calculated from the median WNT reporter fluorescence from 5000 cells and is expressed as percentage of the fold-induction in the absence of heparin to facilitate comparisons. (**B–D**) WNT reporter induction following treatment of the indicated

*Figure 4 continued on next page*

*Figure 4 continued*

cell lines with 2.78% WNT3A CM -/+20 ng/ml untagged RSPO3, except for LGR4/5/6^KO; PIGL^KO and LGR4/5/6^KO; GPC4/6^KO cells, which were treated with 11.1% WNT3A CM -/+20 ng/ml untagged RSPO3 (since depletion of PIGL or of GPC4 and GPC6 reduces signaling at low doses of WNT (*Lebensohn et al., 2016*), different WNT3A CM concentrations were used to achieve comparable signaling responses to WNT3A alone in all cell lines, so that potentiation by RSPO3 could be compared directly). The fold-induction (over WNT3A CM alone) by RSPO3 was calculated from the average of the median WNT reporter fluorescence from duplicate (**C**) or triplicate (**B and D**) wells and is expressed as percentage of the average fold-induction for WT HAP1-7TGP (left two genotypes in **B**) or LGR4/5/6^KO (right two genotypes in **B** and all genotypes in **C** and **D**) cells to facilitate comparisons. Each circle represents a distinct clonal cell line (***Supplementary file 1***) and the average of data from two or three clonal cell lines for each genotype is indicated by a horizontal line. Significance was determined as described in Materials and methods. (**E**) Proposed models for LGR-dependent and LGR-independent signaling by RSPOs. See text for details.

DOI: https://doi.org/10.7554/eLife.33126.007

The following figure supplement is available for figure 4:

**Figure supplement 1.** LGR-independent signaling by RSPO3 requires heparan sulfate proteoglycans.

DOI: https://doi.org/10.7554/eLife.33126.008

embryo and *Rspo2^-/-* mice have defects in both limb development and branching of bronchioles in the developing lung (*Nam et al., 2007a*; *Nam et al., 2007b*; *Bell et al., 2008*). *Rspo3* has been shown to be important for angioblast and vascular development and *Rspo3^-/-* mouse embryos die around day E10 due to defects in placental vascularization (*Aoki et al., 2007*; *Kazanskaya et al., 2008*). While all three *Lgr*s have not been knocked out simultaneously in mice, *Lgr4^-/-*; *Lgr5^-/-* double knock-out embryos have impaired development of the intestine, kidney and skin (*Kinzel et al., 2014*). We speculate that the lung, limbs and vasculature, which depend on *Rspo2* or *Rspo3* but not *Lgr4* and *Lgr5* function, are tissues in which the LGR-independent mode of signaling may be important during development.

While LGRs are specifically expressed in some stem cell compartments (*Clevers et al., 2014*), HSPGs are expressed ubiquitously on the surface of most cells (*Park et al., 2000*). Thus, LGR-mediated RSPO signaling may be critical when high-level WNT signaling must be restricted to specific cells, as in the stem cell niches of various tissues, while HSPG-mediated RSPO signaling might confer amplification of WNT responses more broadly during the development of organs like the limbs, the lungs or the placenta.

In summary, our results define two alternative modes of RSPO-mediated amplification of WNT/β-catenin signaling that differ in their use of either LGRs or HSPGs (*Figure 4E*). The mutant and chimeric RSPO ligands we described should allow the selective modulation of these alternate modes of signaling.

## Note added in proof

We recently became aware of soon-to-be published experiments from Bruno Reversade and colleagues (E. Szenker-ravi et al., 2018) that suggest an LGR-independent role for RSPOs during mouse development.

## Materials and methods

The following Materials and methods relevant to this manuscript have been described previously (*Lebensohn et al., 2016*): cell lines and growth conditions, preparation of WNT3A conditioned medium (CM) and construction of the HAP1-7TGP WNT reporter haploid cell line.

### Reagent providers

Reagents were obtained from the following providers: GE Dharmacon, Lafayette, CO; Addgene, Cambridge, MA; New England Biolabs (NEB), Ipswich, MA; Integrated DNA Technologies (IDT), Inc., Coralville, IA; R&D Systems, Minneapolis, MN; Selleck Chemicals (Selleckchem), Houston, TX; Gemini Bio-Products, West Sacramento, CA; BD Biosciences, San Jose, CA; GE Healthcare Life Sciences, Logan, UT; Polysciences, Inc., Warrington, PA; Thermo Fisher Scientific, Waltham, MA; Sigma-Aldrich, St. Louis, MO; Pall Corporation, Fribourg, Switzerland; Bio-Rad, Hercules, CA; Li-Cor, Lincoln, NE; BioLegend, San Diego, CA.

## Plasmids

pCX-Tev-Fc (unpublished) was a gift from Henry Ho (University of California Davis, Davis, CA). pHLsec-HA-Avi-1D4 (unpublished, derived from pHLSec (*Aricescu et al., 2006*) by incorporating a C-terminal HA tag following the signal sequence and an N-terminal Gly/Ser linker, AviTag biotinylation sequence and 1D4 tag [*Molday and Molday, 2014*]) was a gift from Christian Siebold (University of Oxford, Oxford, United Kingdom). RSPO1-GFP (*Ruffner et al., 2012*) was a gift from Feng Cong (Developmental and Molecular Pathways, Novartis Institutes for Biomedical Research, Cambridge, MA). MGC Human RSPO3 Sequence-Verified cDNA was purchased (GE Dharmacon Cat. # MHS6278-202841214). pX330-U6-Chimeric_BB-CBh-hSpCas9 (pX330) was a gift from Feng Zhang (Addgene plasmid # 42230). pX333 was a gift from Andrea Ventura (Addgene plasmid # 64073).

pHLsec-HA-hRSPO1-Tev-Fc-Avi-1D4 and pHLsec-HA-hRSPO3-Tev-Fc-Avi-1D4 were constructed through a two-step subcloning strategy. In the first step, human RSPO1 and human RSPO3 were amplified by PCR with forward primers pCX-RSPO1-F (5'- GAG GCT AGC ACC ATG CGG CTT GGG CTG TGT G-3') or pCX-RSPO3-F (5'-GAG GCT AGC ACC ATG CAC TTG CGA CTG ATT TCT TG-3'), containing an NheI restriction site, and reverse primers pCX-RSPO1-R (5'-TGA GGT ACC AAG GCA GGC CCT GCA GAT GTG-3') or pCX-RSPO3-R (5'- TGA GGT ACC AAG TGT ACA GTG CTG ACT GAT ACC GA-3'), containing a KpnI restriction site. The products were digested with NheI and KpnI and subcloned into pCX-Tev-Fc digested at the same sites to obtain pCX-hRSPO1-Tev-Fc and pCX-hRSPO3-Tev-Fc.

In the second step, a fragment containing RSPO1 or RSPO3 followed by two tandem Tev cleavage sites, a linker and the Fc domain of human IgG was amplified by PCR from pCX-hRSPO1-Tev-Fc or pCX-hRSPO3-Tev-Fc, respectively, using forward primers pHL-SEC-RSPO1-F-gibson (5'- CGA CGT GCC CGA CTA CGC CAC CGG TAA CCT GAG CCG GGG GAT CAA GGG G-3') or pHL-SEC-RSPO3-F-gibson (5'- CGA CGT GCC CGA CTA CGC CAC CGG TAA CCT GCA AAA CGC CTC CCG GG-3') and reverse primer pHL-SEC-FC-R-gibson-no-KpnI (5'- ACC ACC GGA ACC TCC GGT ACT TTT ACC GGA GAC AGG GGA GA-3'). The forward and reverse primers contained 24 base pair (bp) overhangs complementary to pHLsec-HA-Avi-1D4 upstream of the unique AgeI site and downstream of the unique KpnI site in the vector, respectively. The reverse primer contained a mutation that eliminated the KpnI site in pHLsec-HA-Avi-1D4, hence retaining only one KpnI site between RSPO1 or RSPO3 and the Tev cleavage sites in the resulting construct. The PCR products were subcloned by Gibson assembly (using Gibson Assembly Master Mix, NEB Cat. # E2611L) into pHLsec-HA-Avi-1D4 digested with AgeI and Acc65I (an isoschizomer of KpnI) to produce pHLsec-HA-hRSPO1-Tev-Fc-Avi-1D4 and pHLsec-HA-hRSPO3-Tev-Fc-Avi-1D4, which contain a single AgeI site upstream and a single KpnI site downstream of the RSPO coding sequence. Henceforth, we refer to the vector backbone of this new constructs as pHLsec-HA-Tev-Fc-Avi-1D4.

Human RSPO1 and RSPO3 mutants and chimeras (*Figures 2A* and *3A* and *Supplementary file 2*) were generated synthetically as gBlocks Gene Fragments (IDT), flanked at the 5' and 3' ends, respectively, by 24 bp overhangs overlapping the sequence upstream of the unique AgeI site and downstream of the unique KpnI site in the pHLsec-HA-Tev-Fc-Avi-1D4 vector. The gBlocks were subcloned into pHLsec-HA-Tev-Fc-Avi-1D4, digested with AgeI and KpnI, using the NEBuilder HiFi DNA Assembly Master Mix (NEB Cat. # E2621L).

To remove the dimerizing Fc tag from RSPO3 ΔTSP/BR in order to make the protein used in *Figures 2F and G*, a fragment lacking the TSP and BR domains of RSPO3 was amplified by PCR using forward primer pHL-SEC-RSPO3-F-gibson (sequence described above) and reverse primer pHL-SEC-AVI-1D4-RSPO3FU2-R-gibson (5'-AGA CCG GAA CCA CCG GAA CCT CCG GTA CCC ACA ATA CTG ACA CAC TCC ATA GTA TGG TTG T-3'), containing 24 bp overhangs complementary to pHLsec-HA-Avi-1D4 upstream of the unique AgeI site and downstream of the unique KpnI site in the vector, respectively. The PCR product and pHLsec-HA-Avi-1D4 vector were both digested with AgeI and KpnI and ligated to produce pHLsec-HA-hRSPO3ΔTSP/BR-Avi-1D4.

All constructs were sequenced fully and have been deposited in Addgene.

## Analysis of WNT reporter fluorescence

To measure WNT reporter activity in HAP1-7TGP cells (*Lebensohn et al., 2016*) or derivatives thereof, ~24 hr before treatment cells were seeded in 96-well plates at a density of $1.5 \times 10^4$ per well and grown in 100 µl of complete growth medium (CGM) 2 (*Lebensohn et al., 2016*). Cells were

treated for 20–24 hr with the indicated concentrations of WNT3A CM (*Lebensohn et al., 2016*), untagged recombinant human R-Spondin 1, 2, 3 or 4 (R&D Systems Cat. # 4645-RS, 3266-RS, 3500-RS or 4575-RS, respectively), tagged RSPO1-4 proteins (see below) or CHIR-99021 (CT99021) (Selleckchem Cat. # S2924), all diluted in CGM 2. Cells were washed with 100 µl phosphate buffered saline (PBS), harvested in 30 µl of 0.05% trypsin-EDTA solution (Gemini Bio-Products Cat. # 400–150), resuspended in 120 µl of CGM 2 and EGFP fluorescence was measured immediately by FACS on a BD LSRFortessa cell analyzer (BD Biosciences) using a 488 laser and 505LP, 530/30 BP filters, or on a BD Accuri RUO Special Order System (BD Biosciences).

For the experiments shown in *Figures 1C*, *2B, E*, *3B, E* and *4B–D* and *Figure 4—figure supplement 1A—C*, cells were treated in duplicate or triplicate wells, fluorescence data for 5000–10,000 singlet-gated cells was collected and the median EGFP fluorescence for each well was depicted and/ or used to calculate other parameters depicted, as indicated in the figure legends. For the experiments shown in *Figures 1A,B*, *2C, D, F, G*, *3C, D*, *4A* and *Figure 1—figure supplement 1C and D*, cells were treated in single wells and fluorescence data for 5000–10,000 singlet-gated cells was collected. The median EGFP fluorescence and in some cases the standard error of the median (SEM = 1.253 σ / $\sqrt{n}$, where σ = standard deviation and *n* = sample size) from each well were depicted. Dose-response curves were fitted using the nonlinear regression (curve fit) analysis tool in GraphPad Prism 7 using the [agonist] vs. response – variable slope (four parameters) equation (*Figure 1C*, *2C, D, F, G*, *3C, D* and *Figure 1—figure supplement 1C and D*) or the [inhibitor] vs. response – variable slope (four parameters) equation (*Figure 4A*), both with least squares (ordinary) fit option. Results presented are representative of experiments repeated at least twice.

## Construction of mutant HAP1-7TGP cell lines by CRISPR/Cas9-mediated genome editing

Oligonucleotides encoding single guide RNAs (sgRNAs) (*Supplementary file 3*) were selected from one of two published libraries (*Wang et al., 2015*; *Doench et al., 2016*) or designed using the 'Guide Picker' tool of the DESKGEN Cloud CRISPR design software (https://www.deskgen.com/land-ing/cloud.html). Oligonucleotides were cloned into the single cloning site of pX330 according to a published protocol (*Cong et al., 2013*) (original version of 'Target Sequence Cloning Protocol' from http://www.genome-engineering.org/crispr/wp-content/uploads/2014/05/CRISPR-Reagent-Descrip-tion-Rev20140509.pdf), or sequentially into the two cloning sites of pX333 (*Maddalo et al., 2014*) by digesting the plasmid at each site and ligating the oligonucleotides as described for pX330.

Clonal cell lines derived from HAP1-7TGP were established by transient transfection with pX330 or pX333 containing the desired sgRNAs, followed by single-cell sorting as described previously (*Lebensohn et al., 2016*). Genotyping was done as described previously (*Lebensohn et al., 2016*) using the primers indicated in *Supplementary file 3* and the results are summarized in *Supplementary file 1*. To generate triple, quadruple, quintuple and septuple knock-out (KO) cell lines, a single clonal cell line harboring the first desired mutation or mutations was used in subsequent rounds of transfection with pX330 or pX333 containing additional sgRNAs, followed by single-cell sorting. To facilitate screening of mutant clones by PCR when targeting multiple genes simultaneously, we sometimes targeted one of the genes at two different sites within the same exon or in adjacent exons and amplified genomic sequence encompassing both target sites. Mutant clones were readily identified by the altered size of the resulting amplicon and the precise lesions were confirmed by sequencing the single allele of each gene.

## Production of tagged RSPO proteins by transient transfection of 293T cells and immuno-affinity purification from conditioned media (see *Figure 1—figure supplement 1A*)

~24 hr before transfection, 14 × 10⁶ 293T cells were seeded in each of two T-175 flasks containing 30 ml of CGM 1 (*Lebensohn et al., 2016*) for transfection with each construct. Once they had reached 60–80% confluency, the cells in each flask were transfected with 1 ml of a transfection mix prepared as follows: 22.3 µg of pHLsec-HA-hRSPO-Tev-Fc-Avi-1D4 construct encoding tagged WT, mutant or chimeric RSPO proteins was diluted in 930 µl of serum-free DMEM (GE Healthcare Life Sicences Cat. # SH30081.01) and 70 µl of polyethylenimine (PEI, linear, MW ~25,000, Polysciences, Inc. Cat. # 23966) were added from a 1 mg/ml stock (prepared in sterile water, stored frozen and

equilibrated to 37°C before use). The transfection mix was vortexed briefly, incubated for 15–20 min at room temperature (RT) and added to the cells without replacing the growth medium. ~16 hr post-transfection, the cells were washed with 30 ml PBS and the medium was replaced with 28 ml of CD 293 medium (Thermo Fisher Scientific Cat. # 11913019) supplemented with 1x L-glutamine solution (stabilized, Gemini Bio-Products Cat. # 400–106), 1x penicillin/streptomycin solution (Gemini Bio-Products Cat. # 400–109) and 2 mM valproic acid (Sigma-Aldrich Cat. # P4543, added from a 0.5 M stock prepared in water and sterilized by filtration through a 0.22 µm filter) to promote protein expression.

~90 hr post-transfection, the CM from each of the two flasks, containing secreted RSPO protein, was harvested and centrifuged for 5 min at 400 x g to pellet detached cells. The supernatant was centrifuged for 5 min at 4000 x g and filtered through 0.45 µm filters (Acrodisc syringe filters with Supor membrane, Pall Corporation) to remove particulates and was reserved on ice.

Prior to the purification, Rho 1D4 immuno-affinity resin was prepared by coupling Rho 1D4 purified monoclonal antibody (University of British Columbia, https://uilo.ubc.ca/rho-1d4-antibody) to CNBr-activated sepharose 4B (GE Healthcare Life Sciences Cat. # 17-0430-01). Briefly, 1 g of dry CNBr-activated sepharose 4B was dissolved in 50 ml of 1 mM HCl and allowed to swell. The resin was transferred to an Econo-Pac chromatography column (Biorad Cat. # 7321010) and washed by gravity flow with 50 ml of 1 mM HCl, followed by 50 ml of 0.1 M NaHCO$_3$, 0.5 M NaCl, pH 8.5. 14 mg of Rho 1D4 antibody were dissolved in 0.1 M NaHCO$_3$, 0.5 M NaCl, pH 8.5 and incubated with the resin overnight, rotating at 4°C. The resin was washed with 50 ml of 0.2 M glycine, pH 8.0 and incubated in the same buffer for 2 hr, rotating at RT. The resin was washed sequentially with 50 ml each of: 0.1 M NaHCO$_3$, 0.5 M NaCl, pH 8.5; 0.1 M NaOAc, 0.5 M NaCl, pH 4.0; 0.1 M NaHCO$_3$, 0.5 M NaCl, pH 8.5; PBS, 10 mM NaN$_3$. The packed resin was resuspended in an equal volume of PBS, 10 mM NaN$_3$ to make a ~50% slurry, aliquoted and stored at 4°C.

300 µl of the ~50% slurry of Rho 1D4 resin was added to a 50 ml conical tube containing the RSPO CM and the suspension was incubated 10 hr rocking at 4°C. Following binding and during all subsequent washes the resin was collected by centrifugation for 5 min at 400 x g in a swinging bucket rotor. The beads were washed three times at RT with 25 ml PBS by resuspending the beads in buffer and mixing by inversion for ~1 min. The resin was transferred to a 1.5 ml Eppendorf tube and washed three more times with 1.4 ml of PBS, 10% glycerol.

Following the last wash, the buffer was aspirated and the resin was resuspended in 150 µl of PBS, 10% glycerol to obtain a ~50% slurry. Tagged RSPO protein was eluted by adding 3 µl of a 25 mM stock of 1D4 peptide ((NH$_3$)-T-E-T-S-Q-V-A-P-A-(COOH)) for a final concentration of 250 µM. Elution was carried out by rotating the tube horizontally overnight at 4°C. Following centrifugation of the resin, the eluate was recovered and reserved on ice. The resin was resuspended in 150 µl of PBS, 10% glycerol and 250 µM 1D4 peptide was added. A second round of elution was carried out for 1 hr at RT. Following centrifugation of the resin, the second eluate was recovered and pooled with the first. The final eluate was centrifuged once again to remove residual resin and the supernatant containing tagged RSPO proteins was aliquoted, frozen in liquid nitrogen and stored at −80°C.

## Quantification of tagged RSPO proteins (see *Figure 1—figure supplement 1B*)

4.5 µl and 13.5 µl of the final eluates containing tagged RSPO proteins were diluted with 4x LDS sample buffer (Thermo Fisher Scientific Cat. # NP0007) supplemented with 50 mM *tris*(2-carboxyethyl)phosphine (TCEP), heated for 10 min at 95°C and loaded onto NuPAGE 4–12% Bis-Tris gels (Thermo Fisher Scientific) alongside Precision Plus Protein molecular weight standards (Bio-Rad Cat. # 1610373) and bovine serum albumin (BSA) standards (Thermo Fisher Scientific Cat. # 23209) for quantification. Proteins were electrophoresed in 1x NuPAGE MES SDS running buffer (Thermo Fisher Scientific Cat. # NP0002).

Gels were fixed in 50% methanol, 7% acetic acid for 30 min, rinsed with several changes of water for 1.5 hr, stained with GelCode Blue Stain Reagent (Thermo Fisher Scientific Cat. # 24590) for 2 hr, de-stained in water overnight and imaged using the Li-Cor Odyssey imaging system. Acquisition parameters for coomassie fluorescence (700 nm channel) were set so as to avoid saturated pixels and bands with intensities within the linear range of fluorescence for the BSA standards were quantified using manual background subtraction.

## Immunoblot analysis of tagged RSPO proteins (see *Figure 1—figure supplement 1B*)

50 ng of tagged RSPO proteins were electrophoresed as described above and transferred to nitro-cellulose membranes in a Criterion Blotter apparatus (Bio-Rad Cat. # 1704071) using 1x NuPAGE transfer buffer (Thermo Fisher Scientific Cat. # NP0006) containing 10% methanol. Membranes were blocked with Odyssey Blocking Buffer (Li-Cor Cat. # 927–40000) for 1 hr at RT and incubated over-night at 4°C with purified anti-HA.11 Epitope Tag primary antibody (BioLegend Cat. # 901501; previously Covance cat. # MMS-101P) diluted 1:1500 in blocking solution (a 1 to 1 mixture of Odyssey Blocking Buffer and TBST (Tris buffered saline (TBS), 0.1% Tween-20)). Membranes were washed with TBST, incubated for 1 hr at RT with IRDye 800CW donkey anti-mouse IgG (H + L) (Li-Cor Cat. # 926–32212) diluted 1:10,000 in blocking solution, washed with TBST followed by TBS and imaged using the Li-Cor Odyssey imaging system.

## Preparation of figures and statistical analysis

Illustrations were prepared using PowerPoint (Microsoft) and Illustrator CS6 (Adobe). Bar graphs, dose-response graphs and circle graphs were prepared using Prism 7 (GraphPad Software) and statistical analysis was performed using the same software. For comparisons between two datasets, significance was determined by unpaired t test; for comparisons between more than two datasets, significance was determined by one-way ANOVA. Significance is indicated as **** ($p<0.0001$), ** ($p<0.01$), * ($p<0.05$) or ns (not significant). Pictures of gels and immunoblots were only adjusted for contrast and brightness using Photoshop CS6 (Adobe) and were arranged in Illustrator CS6.

## Data availability

All data generated or analyzed during this study are included in this published article (and its supplementary information files). The RNAseq dataset used in *Table 1* is publicly available (NCBI GEO accession number GSE75515, https://www.ncbi.nlm.nih.gov/geo/).

## Acknowledgements

We thank Jan Carette and Rohatgi lab members for input on the project, Henry Ho for pCX-Tev-Fc, Christian Siebold for pHLsec-HA-Avi-1D4 and Feng Cong for RSPO1-GFP. This work was funded by NIH grants DP2 GM105448 and R35 GM118082 to RR. AML was supported by the Stanford Dean's Postdoctoral Fellowship, the Stanford Cancer Biology Program Training Grant and the Novartis sponsored Fellowship from the Helen Hay Whitney Foundation. The authors declare no conflicting financial interests.

## Additional information

### Funding

| Funder | Grant reference number | Author |
| --- | --- | --- |
| National Institutes of Health | GM105448 | Rajat Rohatgi |
| National Institutes of Health | GM118082 | Rajat Rohatgi |
| Helen Hay Whitney Foundation | Novartis Post-Doctoral Fellowship | Andres M Lebensohn |
| Stanford University School of Medicine | Dean's Post-Doctoral Fellowship | Andres M Lebensohn |
| Stanford University School of Medicine | Cancer Biology Training Grant | Andres M Lebensohn |

The funders had no role in study design, data collection and interpretation, or the decision to submit the work for publication.

## Author contributions
Andres M Lebensohn, Conceptualization, Formal analysis, Investigation, Methodology, Writing—
original draft, Writing—review and editing; Rajat Rohatgi, Conceptualization, Supervision, Funding
acquisition, Methodology, Writing—original draft, Project administration, Writing—review and
editing

## Author ORCIDs
Andres M Lebensohn (iD) https://orcid.org/0000-0002-4224-8819
Rajat Rohatgi (iD) https://orcid.org/0000-0001-7609-8858

## Decision letter and Author response
Decision letter https://doi.org/10.7554/eLife.33126.016
Author response https://doi.org/10.7554/eLife.33126.017

## Additional files

### Supplementary files
• Supplementary file 1. Description of engineered cell lines used in this study. Clonal cell lines
derived from HAP1-7TGP in which a single or multiple genes were targeted using CRISPR/Cas9 are
described in two separate spreadsheets labeled accordingly. When more than one clonal cell line
was generated targeting the same gene or genes, the 'Cell Line Name' column indicates the generic
name used throughout the manuscript to describe the genotype and the 'Clone #' column identifies
distinct individual clones. The figures in which each clone was used are also indicated. The 'CRISPR
guide' column indicates the name of the guide used, which is the same as that of the oligonucleoti-
des encoding sgRNAs (see Materials and methods and *Supplementary file 3*). The 'Genomic
Sequence' column shows 80 nucleotides of genomic sequence (5' relative to the gene is to the left)
surrounding the target site; when two adjacent sites were targeted within the same gene, 80 nucleo-
tides of genomic sequence surrounding each target site are shown and the number of intervening
base pairs that are not shown between the two sites is indicated in parenthesis. Each group of clonal
cell lines made using the same CRISPR guides is separated by a horizontal spacer and the 'Genomic
Sequence' column is headlined by the reference WT genomic sequence (obtained from RefSeq). The
guide sequence is colored blue and the site of the double strand cut made by Cas9 is between the
two underlined bases. Sequencing results for individual clones are indicated below the reference
sequence. Rare differences between the reference RefSeq sequence and the sequencing result
obtained from HAP1-7TGP clones, as well as any undetermined sequences, are indicated in green.
Some clones obtained following transfection with the indicated sgRNAs and single-cell sorting were
found to be WT at the target site or sites; these are indicated as such and were used as controls. For
mutant clones, mutated nucleotides are colored red (dashes represent deleted nucleotides, ellipses
are used to indicate that a deletion continues beyond the 80 nucleotides of sequence shown and
large insertions are indicated in brackets) and the nature of the mutation, the resulting genotype
and any pertinent observations are also described. For clones in which multiple genes were tar-
geted, the CRISPR guide (or pair of guides) used to target each gene as well as the genomic
sequence, mutation, genotype and observations pertaining to each of the targeted genes are desig-
nated '1', '2', '3' and so on in the column headings and are shown under horizontal spacers of differ-
ent colors.
DOI: https://doi.org/10.7554/eLife.33126.009

• Supplementary file 2. Nucleotide sequences of RSPO1 and RSPO3 WT, mutant and chimeric con-
structs used in this study. The name and length of each construct is indicated. Lowercase sequences
overlap the sequence upstream of the unique AgeI site and downstream of the unique KpnI site in
the pHLsec-HA-Tev-Fc-Avi-1D4 vector. Uppercase sequences encode human RSPO1 and
RSPO3 WT, mutant and chimeric constructs. For point mutants, mutated codons are underlined.
DOI: https://doi.org/10.7554/eLife.33126.010

• Supplementary file 3. List of oligonucleotides and primers used to generate and characterize clonal
cell lines engineered using CRISPR/Cas9. The names and sequences of pairs of oligonucleotides
encoding sgRNAs (which were cloned into pX330 or pX333) are shown in the first and second

columns, respectively. The names and sequences of pairs of PCR primers used to amplify corresponding genomic regions flanking sgRNA target sites are shown in the third and fourth columns, respectively. The names and sequences of primers used to sequence the amplified target sites are shown in the fifth and sixth columns, respectively.

DOI: https://doi.org/10.7554/eLife.33126.011

• Transparent reporting form

DOI: https://doi.org/10.7554/eLife.33126.012

## Major datasets

The following previously published dataset was used:

| Author(s) | Year | Dataset title | Dataset URL | Database, license, and accessibility information |
|---|---|---|---|---|
| Dubey R, Lebensohn A, Bahrami-Nejad Z, Marceau C, Sikic BI, Carette J, Rohatgi | 2016 | Gene expression analysis of human haploid cells (HAP1) depleted of SMARCB1 and SMARCA4 | https://www.ncbi.nlm.nih.gov/geo/query/acc.cgi?acc=GSE75515 | Publicly available at the NCBI Gene Expression Omnibus (accession no: GSE75515) |

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
