## [Decision Letter]

Thank you for submitting your manuscript " R-spondins can potentiate WNT signaling without LGR receptors " to *eLife*. Your article has been reviewed by two peer reviewers, and the evaluation has been overseen by a Reviewing Editor (Jeremy Nathans) and a Senior Editor. The reviewers have discussed the reviews with one another and the Reviewing Editor has drafted this letter. Although there appears to be a difference of opinion between the two primary reviewers, I feel the work merits publication in *eLife* and a consensus was reached consistent with this view in the consultation session we convened after the reviews were submitted.

Reviewer #1

This manuscript by Lebensohn and Rohatgi reported that RSPO2 and RSPO3 could potentiate Wnt signaling in the absence of LGR4/5/6 receptors. The authors found that RSPO2 and RSPO3 enhanced the Wnt signaling in a LGR4/5/6 knockout human cell line, whereas the activities of RSPO1 and RSPO4 required the LGR4/5/6 receptors. The authors also used site-directed mutagenesis and domain swapping experiments to reveal that the LGR4/5/6-independent activity of RSPO3 was determined by the N-terminal FU1 domain and C-terminal TSP/BR domains. Finally they proposed a model that in the absence of LGR4/5/6 receptors, the binding of RSPO2 or RSPO3 by cell-surface HSPGs, replacing the binding by LGR4/5/6 receptors, helped to present the bound RSPO to interact with ZNRF3/RNF43, which facilitates the endocytosis and clearance of ZNRF3/RNF43 to increase the Wnt receptor levels on cell surface.

The discovery described in the manuscript is interesting and may help to differentiate four members in the RSPO family in the future research. However, the novelty and content of current research are still limited. Previous studies have shown that RSPO2 Fu1-Fu2 fragment is able to bind and form complex with ZNRF3 without LGR4/5/6 (Zebisch et al., 2013). The interactions between RSPOs and ZNRF3/RNF43 have been clearly elucidated (Zebisch et al., 2013; Chen P et al., 2013). It has also been reported that the TSP/BR domain is involved in interacting with HSPGs, including the report of syndecan 4 as the receptor of RSPO3 (Okhawara B et al., 2011). Furthermore, the physiological importance and relevance of the discovery was not tested in the current study. Indeed, the author mentioned in the last paragraph that the developmental, regenerative and oncogenic context of this discovery need to be defined.

Based on the above stated reasons, I feel that the content and significance of current research are not strong enough for the manuscript to be published in *eLife*.

Reviewer #2

Overall, this work provides significant new insight into the mechanisms of action of R-spondins. The work is very well done from a technical perspective and the work defining the relative roles of syndecans and glypicans added additional important information. The results of this work are likely to be highly influential in the field. I had previously seen this work published on the bioRxiv preprint site and was very intrigued after reading this data there earlier.

I have no requests for additional data to support the work. However, if possible, I would encourage further discussion in the text speculating on where this LGR-independent mode of signaling is most likely to be operative (Final paragraph of Results and Discussion section).

---

## [Author Response]

Reviewer #1[…] The discovery described in the manuscript is interesting and may help to differentiate four members in the RSPO family in the future research. However, the novelty and content of current research are still limited. Previous studies have shown that RSPO2 Fu1-Fu2 fragment is able to bind and form complex with ZNRF3 without LGR4/5/6 (Zebisch et al., 2013). The interactions between RSPOs and ZNRF3/RNF43 have been clearly elucidated (Zebisch et al., 2013; Chen P et al., 2013). It has also been reported that the TSP/BR domain is involved in interacting with HSPGs, including the report of syndecan 4 as the receptor of RSPO3 (Okhawara B et al., 2011). Furthermore, the physiological importance and relevance of the discovery was not tested in the current study. Indeed, the author mentioned in the last paragraph that the developmental, regenerative and oncogenic context of this discovery need to be defined.Based on the above stated reasons, I feel that the content and significance of current research are not strong enough for the manuscript to be published in eLife.

We understand (and explicitly acknowledge in the text) the concern of Reviewer#1 that we have not defined the roles of LGR-independent RSPO signaling in animal or tissue models. We appreciate that a positive consensus was reached regarding the mechanistic insights provided by the work and hope that it incites and facilitates further work to uncover the in vivo roles of LGR-independent signaling.

Reviewer #2Overall, this work provides significant new insight into the mechanisms of action of R-spondins. The work is very well done from a technical perspective and the work defining the relative roles of syndecans and glypicans added additional important information. The results of this work are likely to be highly influential in the field. I had previously seen this work published on the bioRxiv preprint site and was very intrigued after reading this data there earlier.I have no requests for additional data to support the work. However, if possible, I would encourage further discussion in the text speculating on where this LGR-independent mode of signaling is most likely to be operative (Final paragraph of Results and Discussion section).

As requested by Reviewer#2, we have now added a discussion at the end of the paper speculating on potential developmental roles for LGR-independent signaling by RSPO2 and RSPO3. Since the three, partially redundant LGR receptors have not been simultaneously knocked-out in mice, our comments are largely based on the described phenotypes of *Rspo2*-/- and *Rspo3*-/- embryos.

We would like to also call attention to an additional set of data that has been added to Figure 4 (panels C and D), along with supportive data in Figure 4—figure supplement 1. In our original manuscript, we speculated that the requirement for HSPGs might be mediated by syndecans (one of the two major classes of cell-surface HSPGs, the other being the glypicans). This was based on the observations that (1) preventing all HSPG biosynthesis by depleting EXTL3 impaired LGR-independent signaling but (2) preventing glypican biogenesis by depleting PIGL had no effect on LGR-independent signaling. While this manuscript was in review, we have been able to test this hypothesis by simultaneously introducing loss-of-function mutations in the genes (*SDC1-4*) encoding all four syndecans expressed in HAP1 cells (Supplementary file 1). Counter to our initial expectation, eliminating all four syndecans did not impair LGR-independent signaling (Figure 4). As we now describe in the text that accompanies Figure 4, this implies that either syndecans and glypicans are redundant in their ability to support LGR-independent signaling, or that a different HSPG is involved. Despite not being requested by the reviewers, we feel that this additional data is very relevant to the central message of the paper and thus important to include in the revised manuscript. Presenting this data will also prevent unnecessary work by other investigators to dissect the potential role of syndecans (which have already been implicated in RSPO-mediated WNT/PCP signaling).